# Valine-279 Deletion–Mutation on Arginine Vasopressin Receptor 2 Causes Obstruction in G-Protein Binding Site: A Clinical Nephrogenic Diabetes Insipidus Case and Its Sub-Molecular Pathogenic Analysis

**DOI:** 10.3390/biomedicines9030301

**Published:** 2021-03-15

**Authors:** Ming-Chun Chen, Yu-Chao Hsiao, Chun-Chun Chang, Sheng-Feng Pan, Chih-Wen Peng, Ya-Tzu Li, Cheng-Der Liu, Je-Wen Liou, Hao-Jen Hsu

**Affiliations:** 1Department of Pediatrics, Hualien Tzu Chi Hospital, Buddhist Tzu Chi Medical Foundation, Hualien 97004, Taiwan; loveroflois1980@gmail.com (M.-C.C.); u9602041@cmu.edu.tw (Y.-C.H.); 2Department of Pediatrics, School of Medicine, Tzu Chi University, Hualien 97004, Taiwan; 3Department of Laboratory Medicine, Hualien Tzu Chi Hospital, Buddhist Tzu Chi Medical Foundation, Hualien 97004, Taiwan; 101353110@gms.tcu.edu.tw; 4Department of Laboratory Medicine and Biotechnology, College of Medicine, Tzu Chi University, Hualien 97004, Taiwan; 5Department of Biochemistry, School of Medicine, Tzu Chi University, Hualien 97004, Taiwan; 107330102@gms.tcu.edu.tw (S.-F.P.); 107330103@gms.tcu.edu.tw (Y.-T.L.); 6Department of Life Science, College of Science and Engineering, National Dong Hwa University, Hualien 974301, Taiwan; pengcw@gms.ndhu.edu.tw (C.-W.P.); 101726104@gms.tcu.edu.tw (C.-D.L.); 7Department of Life Sciences, College of Medicine, Tzu Chi University, Hualien 97004, Taiwan

**Keywords:** AVPR2, CNDI, V279 deletion–mutation, molecular dynamics simulation, structural analysis

## Abstract

Congenital nephrogenic diabetes insipidus (CNDI) is a genetic disorder caused by mutations in arginine vasopressin receptor 2 (*AVPR2*) or aquaporin 2 genes, rendering collecting duct cells insensitive to the peptide hormone arginine vasopressin stimulation for water reabsorption. This study reports a first identified *AVPR2* mutation in Taiwan and demonstrates our effort to understand the pathogenesis caused by applying computational structural analysis tools. The CNDI condition of an 8-month-old male patient was confirmed according to symptoms, family history, and DNA sequence analysis. The patient was identified to have a valine 279 deletion–mutation in the *AVPR2* gene. Cellular experiments using mutant protein transfected cells revealed that mutated AVPR2 is expressed successfully in cells and localized on cell surfaces. We further analyzed the pathogenesis of the mutation at sub-molecular levels via long-term molecular dynamics (MD) simulations and structural analysis. The MD simulations showed while the structure of the extracellular ligand-binding domain remains unchanged, the mutation alters the direction of dynamic motion of AVPR2 transmembrane helix 6 toward the center of the G-protein binding site, obstructing the binding of G-protein, thus likely disabling downstream signaling. This study demonstrated that the computational approaches can be powerful tools for obtaining valuable information on the pathogenesis induced by mutations in G-protein-coupled receptors. These methods can also be helpful in providing clues on potential therapeutic strategies for CNDI.

## 1. Introduction

Nephrogenic diabetes insipidus (NDI), congenital or acquired, is characterized by failure to concentrate urine despite stimulated by normal or elevated arginine vasopressin (AVP) levels. Congenital nephrogenic diabetes insipidus (CNDI) is a rare disease with an incidence of 4–8 per 1,000,000 live male births [1,2]. Its onset and associated clinical symptoms are similar regardless of the molecular defect. However, the variability of disease severity is reportedly linked to the location of mutations in the incomplete NDI patients [3,4]. In infants, frequent vomiting, hyperthermia, and irritability are initial symptoms of CNDI, which are often confused with gastrointestinal disease, infections, or metabolic disorders [5]. Although polyuria and polydipsia are major symptoms of NDI patients, these symptoms are easily missed by physicians or pediatricians; however, delayed development and failure to thrive would be noted if not adequately treated [4,5].

Arginine vasopressin receptor 2 (AVPR2; OMIM *300538), mainly expressed in the principal cells of renal collecting ducts, is a member of class A G-protein-coupled receptors (GPCRs). AVP released from the posterior pituitary gland is transported to kidneys, where it binds to the AVPR2 receptor, facilitating the coupling of G_s_-protein and stimulating intracellular cyclic adenosine monophosphate (cAMP) production, thereby activating protein kinase A (PKA). Activated PKA phosphorylates aquaporin 2 (AQP2; OMIM *107777), causing AQP2-containing vesicles to fuse with the plasma membrane, allowing water reabsorption through the collecting duct [6]. CNDI is caused by mutations in *AVPR2* or *AQP2*, resulting in AVP unresponsiveness. Most CNDI cases (90%) are caused by *AVPR2* mutations on chromosome Xq28 [7,8]. The remaining 10% are due to *AQP2* mutations and are mostly autosomal recessive [9]. Among *AVPR2* mutations, most mutations responsible for CNDI cause misfolding of the receptor, and a minority are altering catalytic functions [10].

It is accepted that proteins are dynamic entities with internal motions [11]. These motions are decisive in protein conformational changes during receptor-mediated signaling, and they are crucial for receptor functionality [12,13]. Disease-caused gene mutations result in the alterations of protein amino acid sequences, which often alters the internal and external interactions of the proteins, leading to changes in protein motions and conformations, thus affecting the protein functions. In general, GPCRs can undergo dynamic conformation fluctuations during the signal transduction [14]. Molecular dynamics (MD) simulations have been used to investigate the GPCR signaling pathways, activation mechanisms, and to design novel drug candidates that lead to therapeutic advances. For example, a computer-aided method was used for the design of allosteric modulators of two understudied GPCRs, GPR68 and GPR65, by combining homology modeling, molecular docking, and experimental assays [15]. MD simulations were applied to explore a new ligand-binding site of muscarinic acetylcholine receptors, M3 and M4 for drug discovery [16]. All-atom long-term MD simulations were also performed on the opioid receptors to evaluate ligands-induced activation and inhibition mechanisms [17]. Therefore, to obtain the understanding of the effects caused by mutations in detail, sub-molecular evaluations on the mutation-induced alterations of protein dynamics would offer great assistance.

Here, we describe a CNDI case confirmed in our hospital. The V279 deletion–mutation in arginine vasopressin receptor 2 (AVPR2) (AVPR2-∆V279), which caused the disease, is identified in this study, and this is the first reported identification of ∆V279 mutation in Taiwanese CNDI cases. We also demonstrate our effort to analyze the pathogenic consequences of the identified mutation at molecular and sub-molecular levels by using in vitro cellular experiments and bioinformatic approaches, including homology modeling, long-term MD simulations, and structural analysis.

## 2. Materials and Methods

### 2.1. Patient Identification and Sample Collection

The suspected diabetes insipidus (DI) was preliminarily diagnosed according to clinical manifestations, family history, and laboratory data. Sample collection from the proband and his family was approved by the Ethics Committee of the Protection of the Human Subjects Institutional Review Board of the Tzu Chi University and Tzu Chi Hospital (approval no: CR109-06). Written informed consent was obtained from the patient’s parents.

### 2.2. Genomic DNA Sequencing

Whole exon genomic DNA mutation screening of *AVPR2* and *AQP2* was performed via Sanger direct sequencing [2,18]. The primers used for PCR are listed in Appendix A. Briefly, following an automated extraction protocol (Genomic DNA Whole Blood kit, MagCore HF16, RBC Bioscience, New Taipei City, Taiwan), genomic DNA was extracted from peripheral blood leukocytes, quantified on NanoDrop^®^ ND1000 (Thermo Fisher Scientific, Waltham, MA, USA) spectrophotometer. DNA aliquots were stored at −20 °C until use. All target sequences’ exons and flanking intronic regions were amplified by PCR under the following conditions: 60 s at 95 °C, 60 s at 61 °C, and 60 s at 72 °C for 35 cycles. Next, PCR products were separated by electrophoresis in 1.5% agarose gel, purified enzymatically, and sequenced using a Big Dye kit (Applied Biosystems, Beverly, MA, USA) [2].

### 2.3. Plasmid Construction and Immunofluorescence Confocal Microscopy

The Flag-tagged wild-type AVPR2 and AVPR2-∆V279 plasmids were generated by subcloning each target gene’s full-length cDNA into the BamHI *Xho*I sites of a pCMV-3Tag-1A vector (Yao-Hong Biotechnology, New Taipei City, Taiwan). Human embryonic kidney 293T (HEK293T) were cultured in Dulbecco’s Modified Eagle Medium (DMEM) medium supplemented with 10% fetal bovine serum, 2 mM L-glutamine, and penicillin (100 U/mL)/streptomycin (100 μg/mL) (Thermo Fisher Scientific, Waltham, MA, USA). 293T cells (5 × 10^4^) were co-transfected with 2 μg of wild-type or mutant AVPR2 and the CherryPicker Cell Capture plasmids (Takara Bio, Shiga, Japan) using a calcium phosphate transfection procedure [19]. The Flag-tagged epitope-specific mouse monoclonal antibody (Sigma-Aldrich, St. Louis, MO, USA) was used for immune-staining, followed by phosphate buffered saline (PBS) washing. Then, Flag-tagged wild-type or mutant AVPR2 was visualized using a rhodamine-conjugated secondary antibody (Jackson ImmunoResearch, Philadelphia, PA, USA). ProLong^®^ Gold antifade reagent with DAPI (4′,6-diamidino-2-phenylindole) (Life Technologies, Carlsbad, CA, USA) was used for nucleus staining. Fluorescence confocal microscopy was performed using a confocal microscope set, CARV II™ Confocal Imager, equipped with an inverted Olympus IX71S8F3 optical microscope.

### 2.4. Homology Modeling of AVPR2 Structure

Although the structure of the AVP-AVPR2-G_s_ complex in active form was reported very recently based on cryogenic electron microscopy (cryo-EM) maps [20], the inactive AVPR2 structure is still unavailable; therefore, the structures used in this study are based on homology modeling from the GPCR database (https://gpcrdb.org/, accessed on 15 March 2021) using 5-HT_2B_ receptor (PDB: 4IB4 [21]) (sequence similarity 38%, for the intermediate state) and NTS_1_ receptor (PDB: 6OSA [22]) (sequence similarity 40%, for the active state) as templates.

### 2.5. Molecular Dynamics (MD) Simulations

AVPR2 receptors at different states were inserted into a 1-palmitoyl-2-oleoyl-sn-glycero-3-phosphocholine (POPC) lipid bilayer system (2 × 144 lipids) by removing overlapping lipids and water molecules for further MD simulations. Solvated water boxes (9.6 × 9.6 × 9.6 nm^3^) were added with ions (Na^+^ and Cl^−^) to generate 0.15 mol/L NaCl solution for energy minimization. All simulations were carried out with GROMACS-2018 using a GROMOS54A7 force field with an integration step size of 2 fs. Simulations were conducted in the *NPT* ensemble employing the velocity-rescaling thermostat at 310 K and 1 bar. Temperatures of the complexes, lipids, and solvents were separately coupled with a coupling time of 0.1 ps. Semi-isotropic pressure coupling was applied with a coupling time of 0.1 ps and compressibility of 4.5 × 10^−5^·bar^−1^ for the xy-plane and the z-axis. The particle-mesh Ewald (PME) summation algorithm with grid dimensions of 0.12 nm and an interpolation order of 4 was used to calculate long-range electrostatic interactions. Lennard–Jones and short-range Coulomb interactions were cut off at 1.4 and 1.0 nm, respectively. The detailed MD simulation equilibration procedures were in the following based on our previous studies [23,24,25]: (i) temperatures were gradually increased from 100 to 200 K, then to 310 K, and systems were run for 500 ps under each temperature. During these simulations, complex structures remained fully restrained (*k* = 1000 kJ·mol^−1^·nm^−2^); (ii) at 310 K, restraints on the structure via the force constant *k* were gradually released starting from *k* = 500 kJ·mol^−1^·nm^−2^ to 250 kJ·mol^−1^·nm^−2^, and then to 100 kJ·mol^−1^·nm^−2^. Each step was run for 2.0 ns. After equilibration, production runs were carried out without any constraint on these complex structures.

Parameters for all simulations in detail are listed in Appendix A. Two replicates of MD simulations for each AVPR2 system were performed. For MD simulations, GROMACS (http://www.gromacs.org/, accessed on 15 March 2021) and Molecular Operating Environment (MOE) software package MOE2019.01 (http://www.chemcomp.com, accessed on 15 March 2021) were used for visualization and analysis. The calculations of residue distances were performed using GROMACS code g_bond.

## 3. Results

### 3.1. Patient Characteristics

An 8-month-old male infant presenting delayed development and long-term frequent vomiting was admitted to Hualien Tzu Chi Hospital, Taiwan with a preliminary diagnosis of hypernatremia and suspected DI. He was the second child of a non-consanguineous healthy couple. The child was a full-term baby weighing 2800 g (15th percentile) at birth. The newborn screening returned normal results. At 3 months, the patient preferred water to milk and developed frequent non-bilious vomiting after feeding. The patient had light-colored urine, a relatively high body temperature (37 °C), and frequent irritability episodes, which subsided after drinking water.

Due to delayed development, the patient was admitted to the hospital and examined at 8 months. The patient had a bodyweight of 6.2 kg, a body height of 64 cm, and a head circumference of 42 cm (all <3rd percentile). Neurological examination revealed hypotonia in four limbs and hyperreflexia in the bilateral knee jerk. Hypernatremia (Na: 154 mmol/L), high serum osmolarity (313 mOsm/kg), and low urine osmolality (92 mOsm/kg) were detected, suggesting a diagnosis of DI. The patient’s 7-year-old sister was healthy and thriving, yet an uncle was diagnosed with NDI during childhood. The patient’s family pedigree is shown in Figure 1A.

After admission, water deprivation and desmopressin tests were conducted to complement the diagnosis. The results, which are summarized in Table 1, confirmed NDI. Considering his family history of NDI, X-linked hereditary NDI was suspected. Moreover, at that time, the patient’s mother was pregnant with a male fetus. Initially, the patient was treated with hydration for hypernatremia and a thiazide agent for NDI. However, thiazide’s side effects (hypokalemia) were noted, and thiazide was replaced by combination therapy (thiazide and spironolactone). We also prescribed an NSAID agent with ibuprofen for persistent polyuria manifesting after combination therapy. A nutritionist helped establish a low-salt and normal-protein diet to address the patient’s decreased urine output and nutritional status. Symptoms of frequent vomiting gradually subsided after hypernatremia was corrected; however, polyuria was still noted. Follow-up treatment showed that the patient’s serum sodium level was within the normal range and that the failure to thrive and delayed development also improved after discharge.

### 3.2. DNA Sequence Analysis

For molecular diagnosis, DNA sequencing analysis was arranged with blood samples from the patient and his parents, and amniotic fluid analysis of the fetus. All of the coding regions of the *AVPR2* and *AQP2* of the proband were sequenced, revealing a hemizygous 3-base pair in-frame deletion at coding position 279 (c.835_837delGTC, p.Val279del) in the exon 2 of *AVPR2* (Figure 1B), which is reported for the first time in Taiwan. We performed additional sequencing analysis in the proband’s close family members, and the same heterozygous mutation was identified in his mother’s genome (Figure 1C). The V279 deletion was absent in his father and the male fetus (Figure 1D,E). There was no abnormal variation in *AQP2* among the family members screened. The DNA sample of the patient’s uncle was unavailable due to an accident-related, premature death.

### 3.3. Both AVPR2 and AVPR2-∆V279 Can Localize to the Cell Plasma Membrane

To evaluate whether AVPR2-∆V279 retains its capability to traffic and localize on the plasma membrane of cells, Flag-tagged wild-type AVPR2 or AVPR2-∆V279 were transfected into HEK293T cells. HEK293T cells are human kidney cells that do not express endogenous AVPR2, allowing only transfected AVPR2 expression. Representative confocal microscopic images of the wild-type and mutant AVPR2-expressing cells are shown in Figure 2. Both the transfected wild-type and mutant AVPR2 were successfully expressed in the kidney cells, and the mutant AVPR2 could traffic to the cell membrane and co-localize with the membrane marker (a membrane-anchored fluorescent protein m-cherry). This result is similar to that of the wild-type protein, indicating that the underlying pathogenesis of the AVPR2-∆V279 might not be due to the expression disorder, ER retention, or constitutive endocytosis, but the functional anomaly of the mutated receptor.

### 3.4. Conformational Changes in the Distinct Types of AVPR2 during MD Simulations

The functionality of protein is intimately related to protein dynamics and conformational changes. In this study, we applied bioinformatic approaches to understand the cause of the functional loss of AVPR2 by the identified mutation at sub-molecular levels. For bioinformatic analysis of structural alterations in AVPR2 caused by the V279 deletion, three AVPR2 systems from homology modeling, including wild-type (AVPR2-WT), partially active mutant AVPR2-D136A [26], and inactive mutant AVPR2-ΔV279, were used. A previous study has shown that apo AVPR2-WT has constitutive activity and its mutant (AVPR2-D136A) has 2- to 3-fold higher constitutive activity in the absence of agonists [26]. In addition, the outward dynamic movement of the transmembrane helix 6 (TM6), which enlarges the cytoplasmic region of GPCR to facilitate G-protein coupling, is key to GPCR activation [25,27,28,29,30,31].

Figure 3 shows the side and bottom views of the superposition of AVPR2 at various simulation periods (0, 500, 1000, and 1500 ns) for different simulation systems, and Figure 4 shows the superposition of TM6 structures in different AVPR2 systems. The simulation results indicated that the TM6 of AVPR2-WT and AVPR2-D136A moved slightly outward for approximately 0.35 and 0.21 nm, respectively. However, for AVPR2-ΔV279, the TM6 tilted inward for approximately 0.57 nm (Figure 4D). Upon comparison of the final frames of the long-term MD simulations, significant differences in the position of TM6 between AVPR2-ΔV279 and apo AVPR2-WT/apo AVPR2-D136A were observed in the bottom half of the helix (Figure 4A). The V279 deletion results in an inward positional change of the bottom half of TM6 of the receptor. Upon superposition of the structures viewed from the receptor bottom (Figure 4B), the differences between the ΔV279 mutant and WT receptor were apparent. The V279 deletion caused an alteration of TM6 dynamic fluctuation and resulted in the bottom half of the helix to move toward the center of the G-protein binding pocket, which should obstruct G-protein coupling. On the other hand, the superposition showed that the positions of the upper part of TM6 among AVPR2-ΔV279, AVPR2-WT, AVPR2-D136A, and putatively activated AVPR2 are similar (Figure 4A). During the MD simulations, the ΔV279 mutation did not cause large-scale structural variations and dynamic movements of receptor helices on the extracellular side, as compared to the positions of helices of AVPR2 at the initial time, and of the putative activated form of AVPR2 (Figure 4C, top view). The homology model of the activated form of AVPR2 indicates an outward movement of TM6, similar to other activated GPCRs [27,29,30]. The TM6 dynamic fluctuation of the wild-type and partially activated AVPR2 systems showed the same trend. In contrast, the TM6 in the AVPR2-ΔV279 system tilted inward and gave negative distance values (Figure 4D). The opening of the G-protein binding pocket upon GPCR activation can also be indicated by the change in the distance between TM4 and TM6 [27], which is measured with the distance between the C_α_ atoms of N157^4.43^ in TM4 and A264^6.28^ in TM6, and the measurements revealed that the TM4-TM6 distances for AVPR2-WT and AVPR2-D136A systems during simulations gradually increased to 2.7 and 2.5 nm under constitutive activity. However, the distance declined with time to 1.9 nm for the AVPR2-ΔV279 system (Figure 4E), indicating that the mutation causes a conformational change, narrowing the intracellular G-protein binding region of AVPR2, making the G-protein recruitment unfavorable and thus suppressing the downstream signaling.

## 4. Discussion

This study reported the first Taiwanese CNDI patient with a V279 deletion in the *AVPR2* gene. A recent study with long-term follow-up by Sharma et al. revealed that the median age of CNDI diagnosis was 0.6 years, which is in agreement with our patient [32]. The median age of 0.6 years for diagnosis is very often due to the fact that the early severe symptoms of CNDI are very similar to those of other diseases, and they are easily missed by physicians and pediatricians. In this study, we performed a series of diagnostic tests of CNDI, as indicated in Table 1. Among the tests, we did perform a water deprivation test on the patient. It is worth noting that the water deprivation test is potentially dangerous and requires careful monitoring of conditions of the patient. As the baseline plasma sodium level was already higher than 150 mmol/L [33], the water deprivation test is therefore unnecessary. This dehydration test should be excluded in the future diagnosis in similar situations. Sharma et al. also found that regular treatment favorably affects the long-term prognosis of CNDI regarding growth, as well as renal and intellectual function [32]. Early diagnosis and rigorous treatment are essential to improve the long-term growth and prognosis of CNDI patients [34]. Genetic analysis for CNDI is essential in facilitating the early diagnosis and providing the appropriate treatment strategies for these patients [9,35]. Approximately 300 putative NDI-causing *AVPR2* mutations have been identified, most of which are missense mutations [9,36], while deletion mutations account for only 3–4% of NDI cases [35]. Our study revealed a case with an in-frame deletion mutation on V279 of AVPR2. There have been a few reported cases of a V279 deletion on AVPR2 worldwide. The V279 deletion of AVPR2 was firstly described in 1993 in a Japanese family and was reported to the Human Gene Mutation Database [37]. In 1994, a patient in Italy was identified to bear the V279 deletion in AVPR2 [38]. In 1998, another study analyzed the *AVPR2* gene in three Japanese families with X-linked NDI and found a single codon in-frame deletion in two of the families in two consecutive valine residues at positions 278 and 279 [39]. In the same year, a female carrier of ΔV279 was identified in the U.S. [40]. In 2000, large-scale screening of *AVPR2* mutations among 117 families diagnosed with NDI identified three NDI families carrying the ΔV279 mutation, one in the U.S. and two in Canada [1]. Considering that the NDI families with AVPR2-ΔV279 mutation are of diverse ethnic and geographic origins, these previous studies indicated that V279 might be a hot spot of deletion–mutation for AVPR2. Notably, none of the reports above investigated the pathogenic mechanism and only focused on mutation screening. The sequence consensus of V279 is 31%, and hydrophobic residue at this position is 83% among GPCR family members, indicating the importance of the hydrophobicity at this position. The deletion of this residue expectedly results in severe consequences on the receptor structure. Previous studies on *AVPR2* mutations have suggested that the loss-of-function mutations on *AVPR2* can be roughly classified into three categories [10,41]: those that decrease, truncate, or even null the expression of the receptor (class I); those that cause the misfolding of the receptor, resulting in its retention in the ER (class II); and those that cause the disorder in the ligand-binding or G-protein coupling ability of the receptor (class III). Among these classes, class II disorders are the most common and occur in approximately 70% of NDI cases [10]. Nevertheless, in our study, based on fluorescence confocal microscopy, AVPR2-ΔV279 was successfully expressed in transfected cells, and its trafficking and plasma membrane localization were comparable to those observed in wild-type counterparts. This suggests that the identified mutant may be classified as a class III disorder.

Owing to advancements in computer technology, in silico methods have become powerful tools for biomedical research. To understand the detailed effects caused by V279 deletion on the structure of AVPR2, we performed comprehensive computational analyses. The GPCRs contain a transmembrane domain consisting of seven transmembrane helices. Many disease-causing mutations occur in this domain as compared with extracellular or intracellular domains [42]. The V279 is located at TM6 and is highly conserved among mammalian AVPR2 receptors, indicating its importance for AVPR2 function [7,37,43]. The crystal structure of apo AVPR2 has not yet been elucidated; however, the homology model of the AVPR2 structure from the GPCR database enables us to perform atomic-level dynamic structural analysis. Indeed, in silico analysis did provide valuable information to understand the effect caused by the mutation. While there was not much difference between the structures of AVPR2-ΔV279 and AVPR2-WT on the extracellular side (Figure 4C), the deletion of V279 in TM6 induced an inward dynamic movement on the intracellular side toward the center of the G-protein binding pocket (Figure 3 and Figure 4A,B). Thus, we hypothesize that the functional loss of AVPR2 caused by the V279 deletion is due to G-protein coupling issues rather than the effect on ligand-binding.

Moreover, homology modeling for different AVPR2–G_s_ complex structures using A_2A_AR-G_s_ as a template [44] indicated that activated AVPR2 could accommodate G_s_-protein coupling without any steric hindrance due to the outward dynamic movement of TM6, whereas mutant AVPR2-ΔV279 was unable to bind the G_s_-protein due to steric repulsion with the α5-helix of G_s_-protein caused by the inward tilt of TM6 (Figure 5). This and previous results [37] are consistent with our MD simulations. Similar findings were confirmed by mutagenesis experiments and MD simulations, suggesting that the mutations at L244^6.40^P and L246^6.42^P of the chemokine receptor CXCR4 do not affect its CXCL12 ligand-binding but noticeably reduce the calcium flux, leading to CXCR4 inactivation [25,45,46]. Very recently, Wang et al. reported a cryo-EM structure of AVP-AVPR2-G_s_ complex [20]. As the AVPR2 structure in this complex represents an active conformation, and the structure of inactive AVPR2 was still not available, they compared their structure with inactive OXTR (47% sequence identity in the TMs with AVPR2). Interestingly, they also suggested an outward movement of TM6 for the receptor activation. In addition, the involvement of Y280^6.44^ (the amino acid next to V279) of the AVPR2 was suggested to be important for the AVPR2 activation. However, they also found that the Y280F mutant did not affect AVP binding [20]. The reason can be implied by our study that the mutations in this region might affect G-protein binding rather than the binding of agonist AVP.

The current management options for CNDI, including treatment for dehydration and hypernatremia, thiazide diuretics, prostaglandin synthesis inhibitors, and a low-salt diet, are only partially effective. Although various investigational therapeutic strategies for CNDI, such as the bypass of defective AVPR2 signaling or the rescue of AVPR2 mutants by chemical chaperones, have been proposed [47], the development of new molecules for binding with the mutated receptor to induce allosteric effect and rescue AVPR2 signaling cascade is another promising therapeutic strategy. However, this largely relies on the understanding of the pathogenic effects of the mutant receptor. Overall, this study reports the first identified Taiwanese CNDI case caused by a V279 deletion in AVPR2. We also demonstrate a promising and cost-effective combination of methods to investigate the effects of a mutation on the GPCR structure and the pathogenic mechanisms underlying GPCR mutations.

## Figures and Tables

**Figure 1 biomedicines-09-00301-f001:**
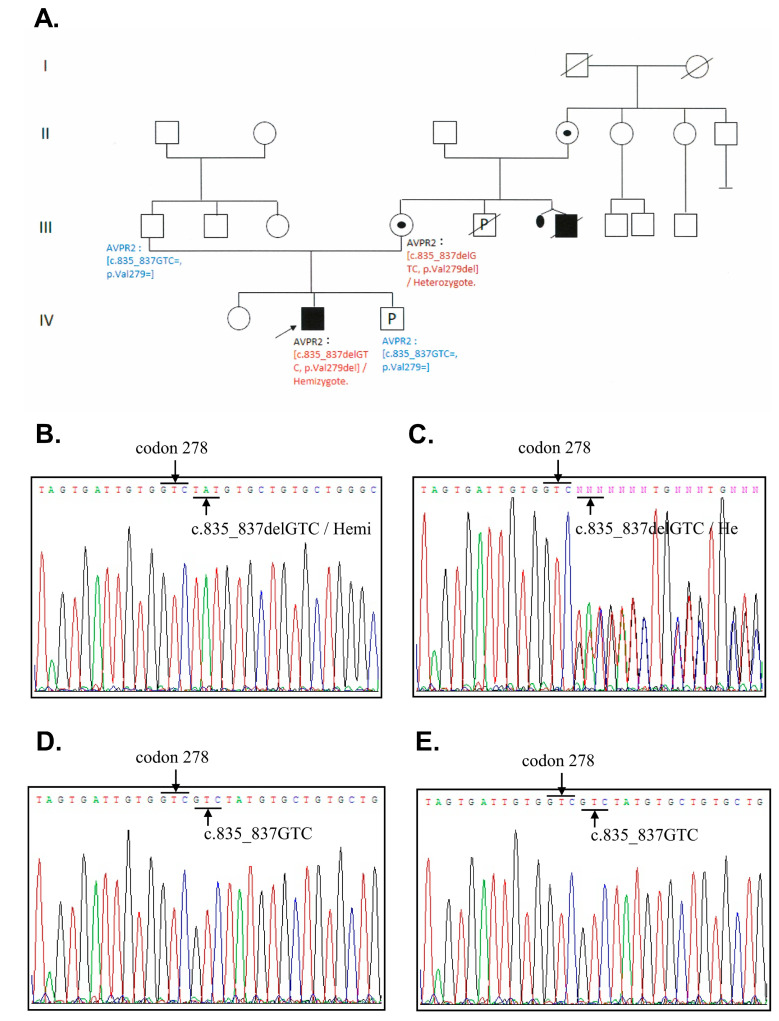
Pedigree of congenital nephrogenic diabetes insipidus and DNA sequencing analyses in the patient’s family. (**A**) Black and white symbols represent clinically affected and unaffected individuals, respectively. The arrow indicates the proband. (**B**–**E**) Arrows represent the area of codon 278 and 279. (**B**) The patient’s DNA sequencing results revealed a hemizygous mutation with a deletion at codon 279 in exon 2 (c.835_837delGTC, p.Val279del). The codon next to codon 278 (GTC) is TAT as indicated by the underline. There should be an additional codon GTC (Val) at the position 279 in between GTC at codon 278 and the TAT (should be at codon 280). (**C**) The patient’s mother revealed a heterozygous mutation with a deletion mutation at codon 279 in exon 2 (c.835_837delGTC, p.Val279del). (**D**) That of the patient’s father showed normal at codon 279 in exon 2. (**E**) That of the fetus showed normal at codon 279 in exon 2.

**Figure 2 biomedicines-09-00301-f002:**
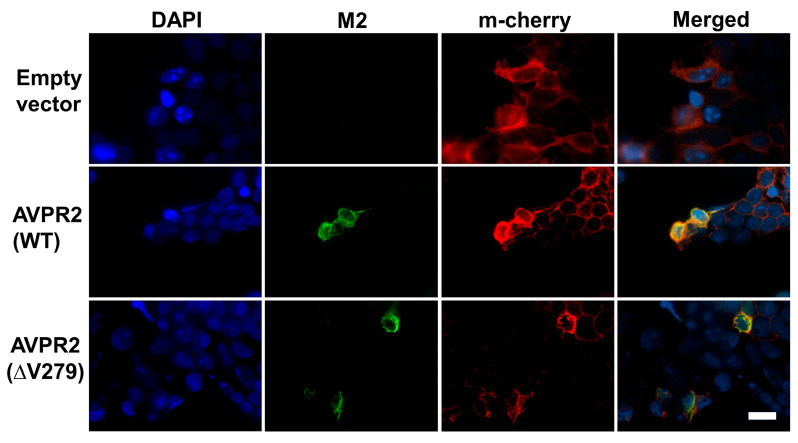
Arginine vasopressin receptor 2 (AVPR2)-∆V279 is localized in the membrane surface. The Flag-epitope tagged AVPR2 or AVPR2-∆V279 expression vector was co-transfected with a membrane-bound cherry fluorescent protein (m-Cherry) vector into HEK 293T cells. M2 antibody was used for immune-staining, and the locations of AVPR2 proteins were visualized by rhodamine-conjugated secondary antibodies. m-Cherry marked the plasma membrane are shown red, and the AVPR2 proteins are in green. Cell nuclei were counterstained by DAPI (blue). Both single staining and merged images are shown. Both the transfected wild-type and ΔV279 mutant AVPR2 proteins were successfully expressed in HEK 293T cells, and both of the proteins can associate with the plasma membrane as indicated by m-cherry co-localization (yellow) in the merged images. The scale bar in the image represents 100 μm.

**Figure 3 biomedicines-09-00301-f003:**
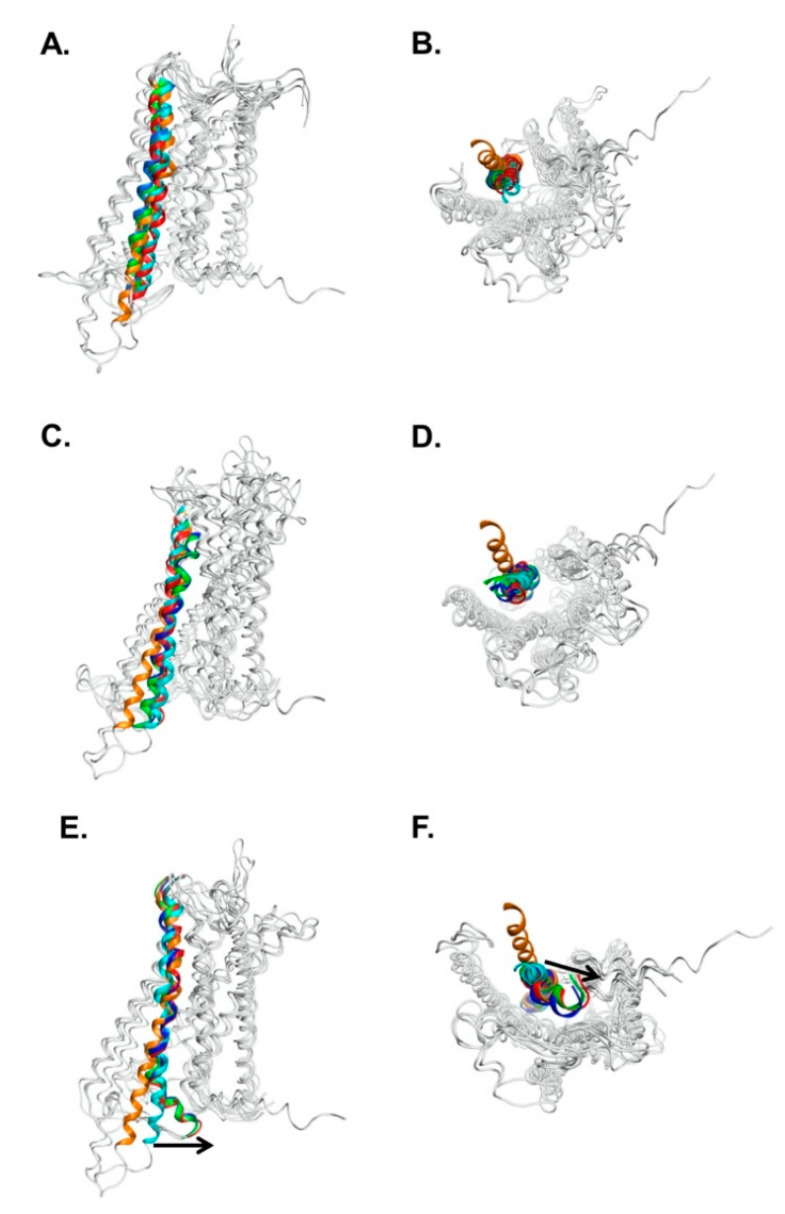
Superposition of structures at different simulation time frames for different AVPR2 systems. (**A**,**B**) Wild-type receptor AVPR2-WT; (**C**,**D**) Partial active mutant receptor AVPR2-D136A; (**E**,**F**) Inactive deletion-mutant AVPR2-ΔV279. (**A**,**C**,**E**) are side views, and (**B**,**D**,**F**) are bottom views. For clarity, only transmembrane helix 6 (TM6) at different time frames is colored, while other regions are shown in gray. Structures in cyan, red, green, and blue indicate the TM6 in the receptor structures at initial time, 500, 1000, and 1500 ns in the MD simulations, respectively; that in orange is TM6 in putative activated form of AVPR2. In AVPR2-WT and AVPR2-D136A, TM6 was tilted slightly outward. However, in AVPR2-ΔV279, the intracellular part of TM6 was shifted inwards, obstructing the G-protein binding site. The arrows indicate the inward movement of TM6.

**Figure 4 biomedicines-09-00301-f004:**
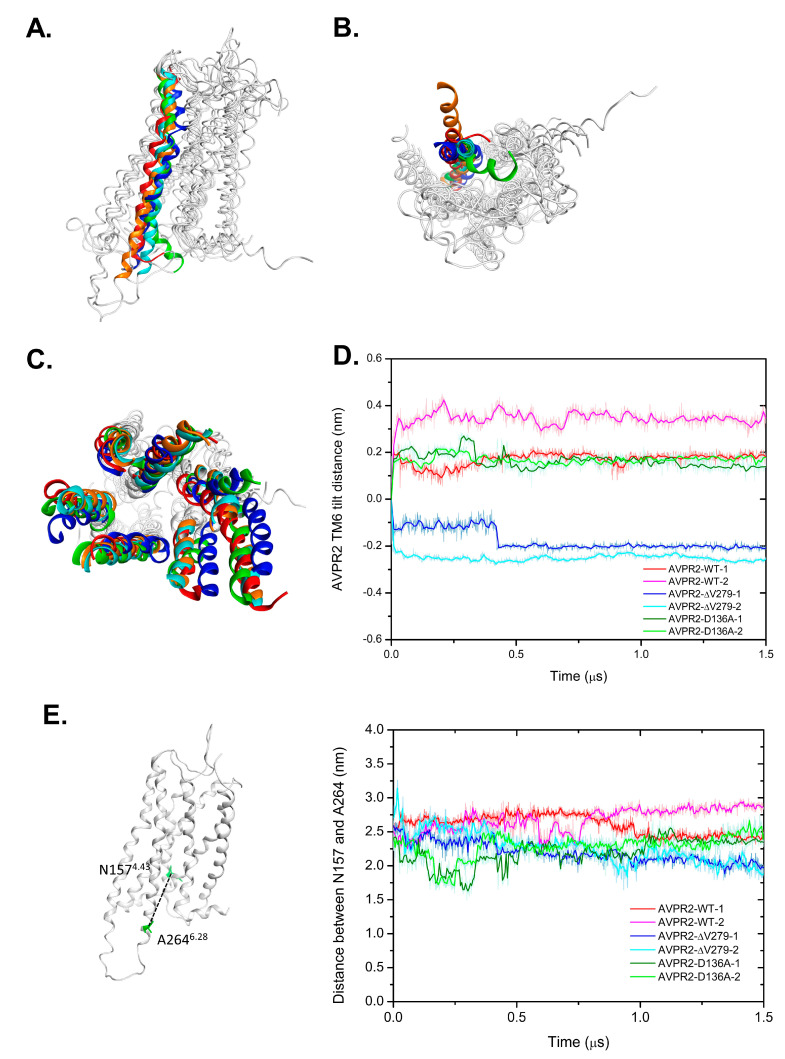
Superposition of TM6 structures in different AVPR2 systems. (**A**) Side view and (**B**) bottom view of the superposition of different AVPR2 systems at final simulation time (1500 ns). Only TM6s at different AVPR2 systems are colored; other regions are in gray. Cyan indicates the initial position of the helix in the simulations; TM6 in red, blue, and green are the positions in AVPR2-WT, AVPR2-D136A, and AVPR2-ΔV279, respectively; TM6 in orange corresponds to the putative activated form. (**C**) The superposition of the extracellular region of different time frames of AVPR2-ΔV279 system (top view). Cyan color: initial time; red color: 500 ns; green color: 1000 ns; blue color: 1500 ns; orange color: active form. (**D**) The change in the tilt distance of TM6 of AVPR2 with time. The C_α_ atom of A264 of TM6 was selected for measurement. (**E**) The profile of transmembrane helices distance with time for different AVPR2 systems. The distance between the C_α_ atoms of A264 of TM6 and N157 of TM4 for each AVPR2 system was measured with time. The dotted line represents the distance between A264 and N157. In (**D**,**E**), two replicates of MD simulations are shown for each AVPR2 system.

**Figure 5 biomedicines-09-00301-f005:**
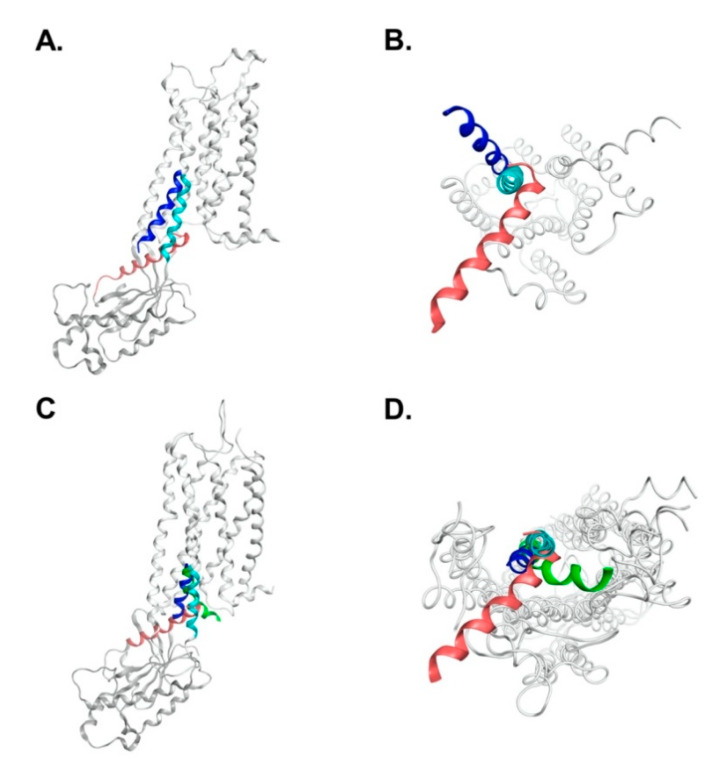
Homology modeled complex structures for A_2A_AR-G_αs_ and AVPR2-G_αs_ systems. (**A**) Side view and (**B**) bottom view of A_2A_AR complexed with the Ras domain of G_αs_. For A_2A_AR, cyan: TM6 at inactive state; blue: TM6 at active state. For Ras-G_αs_, α5-helix: light red. In the inactive state of A_2A_AR, the TM6 may have steric repulsion with α5-helix to inhibit G_αs_ binding. (**C**) Side view and (**D**) bottom view of AVPR2 complexed with the Ras domain of G_αs_. For AVPR2, cyan: initial time; blue: 1500 ns; green: AVPR2-ΔV279. For Ras-G_αs_, α5-helix: light red. α5-helix of G_αs_-protein may have strong steric repulsion with TM6 of the inactive AVPR2-ΔV279.

**Table 1 biomedicines-09-00301-t001:** Results of the water deprivation and desmopressin challenge tests.

Test Duration (h)	Body Weight (g)	Body Weight Loss (%)	Serum Sodium (mmol/L)	Serum Osmolarity (mOsm/kg)	Urine Specific Gravity	Urine Sodium (mmol/L)	Urine Osmolarity (mOsm/kg)
0	6424	0	156	327	1.010	14	152
1	6321	1.6	156	322	1.007	ND	193
2 *	6224	3.1	160	328	1.005	8	128
3	6294	2.0	155	ND	1.006	18	168
4	6156	4.2	160	330	1.008	21	210

* One puff of desmopressin intranasal spray was administered. ND: not done.

## Data Availability

Data supporting our conclusion are presented within this article and the Appendix A.

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
