# Peer review of "Valine-279 Deletion–Mutation on Arginine Vasopressin Receptor 2 Causes Obstruction in G-Protein Binding Site: A Clinical Nephrogenic Diabetes Insipidus Case and Its Sub-Molecular Pathogenic Analysis"

_biomedicines, 2021, doi:10.3390/biomedicines9030301_

Round 1

Reviewer 1 Report

Abstract. The first sentence is inaccurate since, in nephrogenic diabetes insipidus secondary to AQP2 loss-of-function mutations, the principal cells are still responding to AVP by increasing sodium reabsorption (Vasopressin-V2 receptor stimulation reduces sodium excretion in healthy humans. Bankir L, Fernandes S, Bardoux P, Bouby N, Bichet DG.J Am Soc Nephrol. 2005 Jul;16(7):1920-8.).

If the valine 279 AVPR2 deletion is altering signaling, there should be a decreased in cAMP production in transfected cells expressing this mutation, this is not demonstrated here.

Could you give examples in the GPCR field where molecular dynamics stimulation led to therapeutic advances?

Introduction. Ref 1 did not do any prevalence studies but used data from ref 32. Ref 2: there is little variability in the disease severity since most patients have severe signs and symptoms during the first week of life, see recent reference with discussion on mild phenotypes (GENETICS IN ENDOCRINOLOGY Pathophysiology, diagnosis and treatment of familial nephrogenic diabetes insipidus. Bichet DG.Eur J Endocrinol. 2020 Aug;183(2):R29-R40.).

“polyuria and polydipsia not significant in infants with CNDI” this is false.

Third paragraph of the introduction: most AVPR2 mutations responsible for CNDI are misfolded and a minority are altering catalytic components.

Fourth paragraph: “ the first mutation identified in Taiwan” this is an empty statement since the prevalence of mutations is constant in various ancestral groups, they are other AVPR2 mutations not yet reported in Taiwan.

Results. The mutation screening consists of what?

Fig 1 part B: the line under TAT where you indicate the c.835_837 del GTC is confusing since TAT is the next intact a.a.. This should be explained in the legend.

Table 1: you should indicate that a water deprivation test was not necessary here, futile and potentially dangerous, since the baseline plasma sodium was already higher than 150mmol/l, that is, a maximal stimulation of endogenous AVP (Fenske W, Refardt J, Chifu I, Schnyder I, Winzeler B, Drummond J, Ribeiro-Oliveira Jr A, Drescher T, Bilz S, Vogt DR et al. A copeptin-based approach in the diagnosis of diabetes insipidus. New England Journal of Medicine 2018 379 428–439.).I am very firm on this point since dehydration tests are unnecessary in patients with plasma sodium higher than 150 mmol/l.

In fig 2, are you using the same amount of transfected protein to express the wild-type and the deletion mutation on the cell membrane?

Page 7: explain “apo” in apoAVPR2 wild type. Is AVPR2-D136A identified in humans and inducing a gain-of-function with hyponatremia?

Discussion: a median age of 0.6 years for diagnosis means that the early severe manifestations of polyuria and polydipsia are missed by most physicians and pediatricians.

The deltaV279 is responsible for 3-4% of NDI cases: please quote these publications, you need 12 male cases for 300 mutations.

Reviewer 2 Report

General Comment: This is an interesting paper reporting a patient with congenital NDI due to a valine-279 mutation in the V2R.  This mutation has been previously reported in several countries but this is the first report from Taiwan.  The authors provide a molecular dynamic simulation that suggests a hypothetical mechanism by which this mutation results in the disease phenotype.

Major Comments:

  1. The authors hypothesize that the mutation results in suppression of downstream signaling.  It would seem that the authors could test their hypothesis using their HEK-293T cells and determine whether the mutation interferes with cAMP production and/or PKA activation.  This would strengthen the paper.
  2. Abstract line 24 - the V2R and AQP2 are expressed along the entire collecting duct, not just the cortical collecting duct.

Minor Comment: Line 46 - simulated should be simulation

Author Response

The authors hypothesize that the mutation results in suppression of downstream signaling.  It would seem that the authors could test their hypothesis using their HEK-293T cells and determine whether the mutation interferes with cAMP production and/or PKA activation.  This would strengthen the paper.

Response :

The authors thank for reviewer’s suggestion. Reviewer’s opinions are well taken by us and we are now working on the set up of a DV279-based cell model in order to precisely characterize the biological phenotypes of the AVPR2 mutant. To monitor the function of AVPR2-DV279 in mediating downstream signaling cascades upon AVP challenge, we are now establishing the stable clones of HEK293 expressing AVPR2 and DV279, respectively. The above cell lines will be used to verify the defects of AVPR2-DV279 in membrane redistribution and cAMP production, and all downstream signaling events, including arrestin recruitment and internalization, PKA activation, AQP2 activation and redistribution. It is expected that at least 8-12 months are required to complete the above-mentioned research work. We are going to present the data from this ongoing project into our next publication. According to preliminary results on cAMP measurements, cAMP production was greatly reduced in AVPR2-DV279 transfected cells upon stimulation by AVP or dDAVP, as compared to the that in wild-type receptor transfected ones. However, these results should be published together with other cellular experiment results. In this present study, the purpose of the initial cell experiments demonstrated is for proving that the mutant receptor is actually on the cell surface, thus the computational and structural studies presented in this manuscript is meaningful.

Abstract line 24 - the V2R and AQP2 are expressed along the entire collecting duct, not just the cortical collecting duct.

Response:

The authors thank for reviewer’s suggestion. We have revised and taken out “cortical” in the sentence in the Abstract section to be more precise.

Line 46 - stimulated should be stimulation

Response:

The authors thank for reviewer’s comment. We replace the word “stimulated” with “stimulation” in the revised manuscript.

Round 2

Reviewer 1 Report

The recent published  EM cryo data of AVPR2 must be included. see appended
